# Utilizing an Endogenous Progesterone Receptor Reporter Gene for Drug Screening and Mechanistic Study in Endometrial Cancer

**DOI:** 10.3390/cancers14194883

**Published:** 2022-10-06

**Authors:** Yiyang Li, Wei Zhou, Xiangbing Meng, Sarina D. Murray, Long Li, Abby Fronk, Vanessa J. Lazaro-Camp, Kuo-kuang Wen, Meng Wu, Adam Dupuy, Kimberly K. Leslie, Shujie Yang

**Affiliations:** 1Department of Obstetrics and Gynecology, The University of Iowa, Iowa City, IA 52242, USA; 2Department of Pathology, The University of Iowa, Iowa City, IA 52242, USA; 3Holden Comprehensive Cancer Center, University of Iowa, Iowa City, IA 52242, USA; 4High Throughput Screening Facility at University of Iowa (UIHTS), Iowa City, IA 52242, USA or or; 5Division of Medicinal and Natural Products Chemistry, Department of Pharmaceutical Sciences and Experimental Therapeutics, College of Pharmacy, The Ohio State University, Columbus, OH 43210, USA; 6Department of Anatomy and Cell Biology, The University of Iowa, Iowa City, IA 52242, USA

**Keywords:** progesterone receptor, mCherry, reporter gene, endometrial cancer, high throughput screening, GeCKO library, HDAC

## Abstract

**Simple Summary:**

In this study, we generate an innovative endogenous progesterone receptor (PR) reporter gene containing endometrial cancer cell lines and use this tool to visualize PR expression in real-time. We also demonstrated two strategies of using this reporter gene: (1). Drug discovery—screening small molecule inducers for PR expression; (2). Mechanistic investigation—screening potential PR repressors using genome-wide CRISPR knockout library.

**Abstract:**

Expression of progesterone receptor (PR) is a favorable prognostic marker for multiple solid tumors. However, PR expression is reduced or lost in malignant tumors. Thus, monitoring and restoring functional PR expression is important in order to sensitize tumor cells to progesterone therapy in endometrial cancer. We developed stable PR reporter gene containing endometrial cancer cell lines monitoring the endogenous PR expression by inserting mCherry and hygromycin resistant gene at the endogenous PR gene locus by CRISPR/Cas9-mediated genome editing technique. This allows efficient, real-time monitoring of PR expression in its native epigenetic landscape. Reporter gene expression faithfully reflects and amplifies PR expression following treatment with drugs known to induce PR expression. Small molecular PR inducers have been identified from the FDA-approved 1018 drug library and tested for their ability to restore PR expression. Additionally, several candidate PR repressors have been identified by screening the genome-wide CRISPR knockout (GeCKO) library. This novel endogenous PR reporter gene system facilitates the discovery of a new treatment strategy to enhance PR expression and further sensitize progestin therapy in endometrial cancer. These tools provide a systematic, unbiased approach for monitoring target gene expression, allowing for novel drug discovery and mechanistic exploration.

## 1. Introduction

The progesterone receptor (PR) is a multifunctional molecule with a critical role in development. PR is also implicated as a tumor suppressor or as a tumor promoter, depending upon the context, in various cancers, including breast, endometrial, and ovarian cancer [1,2]. It is reported that high PR expression is associated with better clinical outcomes [3,4,5,6,7]. However, PR expression is reduced or lost in many malignant tumors, including endometrial cancer, hindering the response to progesterone therapy [3,4].

Endometrial cancer is the most common gynecological malignancy, with incidences (~61,880 new cases/year) and deaths (~12,160 deaths/year) on the rise [8,9]. Progesterone therapy has been used for over 70 years to treat endometrial cancers with relatively good outcomes for primary disease, but with less promising results for metastatic and recurrent disease [4]. Thus, there is a critical need (1) to understand the mechanisms that inhibit PR expression as endometrial cancer progresses and (2) to restore functional PR expression in order to sensitize tumor cells to progestin therapy.

Several different mechanisms have been reported to explain the reduced expression of PR in breast and endometrial cancer, including ligand-dependent downregulation [10,11,12,13], miRNA-mediated post-transcriptional suppression [14,15,16,17], hyperactive Akt signaling [18,19,20], and epigenetic factors [21,22,23,24]. Our group has systematically dissected the mechanisms by which PR is lost and demonstrated that PR expression can be downregulated at distinct molecular levels during disease progression [24]. We demonstrated that PR expression can be downregulated by epigenetic modulation through the polycomb-repressor complex (PRC2) and DNA methylation; histone deacetylase inhibitors (HDACi) and hypomethylating agents can reverse these PR silencing mechanisms. In addition, we demonstrated that progestin effectiveness can be significantly improved when HDACi are combined with progestin therapy [24,25], leading to the approval of a new NIH NCTN trial, NRG-GY011. Translational studies are underway to evaluate the effectiveness of this treatment regimen. We hypothesize that PR expression and activity can be induced by additional small molecular drugs to maximize clinical interventions. Therefore, our objectives are: (1) to explore novel PR inducers from the FDA approved drug library; and (2) to discover potential PR repressors to reveal precise PR downregulation mechanisms. To achieve these objectives, a novel endogenous PR reporter gene is needed to efficiently monitor endogenous PR expression.

To visualize PR expression, we developed an endogenous PR reporter gene. Traditionally, the PR reporter gene is constructed with the progesterone response element (PRE) followed by luciferase [26,27,28]. This exogenous PRE-luciferase PR reporter gene has shortcomings: (1) it is overexpressed thousands of times more than the endogenous PRE-containing PR target gene; (2) it excludes PR activity by binding to the SP1 response element; (3) it cannot reflect endogenous PR epigenetic landscaping. Our reporter gene, however, is superior to the PRE-luciferase PR reporter gene, and allows efficient, real-time monitoring of PR expression in its native epigenetic landscape. This reporter gene is composed of hygromycin as a negative selection marker and mCherry as a positive selection marker. CRISPR/Cas9 genome editing was used to generate a double strand break (DSB) at the -NGG protospacer adjacent motif (PAM) sequence near the end of the *PGR* gene, allowing the donor vector to be inserted with homologous recombination.

The capabilities of our reporter gene include identifying novel target gene modulators with high throughput screening of drug libraries and screening candidate target gene repressors or enhancers using a knockout or knockdown gene screening library. From our studies, we have identified and validated novel drugs able to induce functional PR expression, including romidepsin and CUDC907 from the FDA-approved 1018 drug library. Potential PR repressors contributing to PR suppression, including GLUD2, APH1A, and SGPP2, have been identified through GeCKO library screening and were validated in vitro. This approach, creating stable cells with an endogenous PR reporter gene, was successfully employed in endometrial cancer cells as reported herein; furthermore, the same strategy can be used to study other hormone-driven tumors, such as ovarian and breast cancers in the future.

## 2. Materials and Methods

### 2.1. Cell Lines and Cell Culture

Ishikawa H is an endometrial cancer cell line, gifted from Dr. Erlio Gurpide (New York University). ECC1, an endometrial cancer cell line, was purchased from ATCC. This cell line was reported as a derivative of Ishikawa cells [29,30]. All cell lines have been authenticated using STR analysis by BioSynthesis.

### 2.2. Antibodies and Reagents

Romidepsin (Celgene, Summit, NJ, USA), LBH589 (Novartis, East Hanover, NJ, USA) and PXD101 (Onxeo former Topotarget, Cambridge, MA, USA) were resuspended in DMSO. Carfilzomib (PR-171), Ponatinib (AP24534), Dronedarone HCl, Paroxetine HCl, Bazedoxifene HCl, Benzethonium chloride, CUDC-907, AR-42, SGI-1027, UNC0642, Roxadustat FG-4592, Ruxolitinib, UNC1215, RG2833, Camptothecin, Teniposide, Duloxetine HCl, and Lapatinib were purchased from Selleck Chem (Houston, TX, USA). Alexidine hydrochloride was purchased from Cayman (Ann Arbor, MI, USA). Antibodies against PRA/B (#3153), PRB (#3157), FOXO1 (#2880), p21 (#2947), cyclin D1 (#2926), and Myc (#13987) were from Cell Signaling (Danvers, MA, USA), and mCherry (EPR20579) was from Abcam (Waltham, MA, USA). PAEP (PA5-54152) were from Fisher scientific (Waltham, MA, USA). β-actin antibody (#A1978) was obtained from Sigma Aldrich (St. Louis, MO, USA).

### 2.3. Generation of the Double Strand Break at Exon 8 of PGR

The TGG sequence was identified at exon 8, 29 nucleotides before TGA stop codon using CRISPR sgRNA design website (https://quiltdata.com/grna-search/, accessed on 29 November 2016) followed by the sgRNA sequence: ACCCAAGATATTGGCAGGGA. Double strand sgRNA was synthesized and inserted into LentiCRISPR v1 vector (gift from Dr. Feng Zhang lab, Addgene plasmid #70662, Watertown, MA, USA) by following the instructions. Sanger DNA sequence confirmed the sgRNA in lentiCRISPR v1 vector. The CRISPR-Cas9-sgRNA vector was transfected into Ishikawa and ECC1 cells.

### 2.4. Alt-R Genome Editing Detection Assay

The CRISPR-Cas9-sgRNA transfected cells were selected using 2 µg/mL puromycin, followed by genomic DNA extraction. PCR was conducted to amplify the target sgRNA sequence and the adjacent sequence, followed by Alt-R genome editing detection assay (cat# 1075931, IDT), per the manufacturer’s instructions. Briefly, the PCR products from the wildtype PGR were mixed with the cut PGR to form the heteroduplexes for T7 Endonuclease I (T7EI) digestion. The digested, mismatched PCR products were separated by agarose gel electrophoresis.

### 2.5. Construction of Hygromycin-mCherry Homology Donor Vector

The homologous recombination arm was retrieved using the PGR DNA sequence from the UCSC genome browser (Chr 11q22.1). The T2A sequence was derived from the Ac5-STABLE2-neo vector. The hygromycin sequence was derived from the pQCXIH CMV vector (gift from Dr. Michael Henry) and the mCherry sequence was derived from pt2-efioc-mcherry (gift from Dr. Adam Dupuy). The PR reporter gene (2716 bp) was synthesized, subcloned into the pUC57 vector, and fully sequenced by Abm (Richmond, BC, Canada). Next, the PR reporter gene was subcloned into the PL253-TK vector (gift from Dr. Adam Dupuy) at the NotI and BamHI digestion sites.

### 2.6. Co-Transfections and Validation of the Correct Clones

Equal amounts of the CRISPR-Cas9-sgRNA vector and PR reporter gene vector were co-transfected into Ishikawa and ECC1 cells using Lipofectamine 2000 (Invitrogen, cat #11668-027, Waltham, MA, USA). The transfectants were selected using puromycin (2 µg/mL) and ganciclovir (20 µM) for over one week. Single colonies were grown, and genomic DNA was extracted, followed by junction PCR to select the correctly inserted clones.

### 2.7. Monitoring PR Expression Using High Content Imaging System

Ishikawa clone 12-A2 cells, stably expressing the PR reporter gene, were grown into a 3D-spheroid using a 384-well plate (Corning spheroid microplates, CLS4516m, Glendale, AZ, USA) with 2000 cells in each well, followed by the treatment with the positive control, romidepsin, or FDA approved drugs (in triplicate) at 1 µM in DMSO for 72 h. mCherry signals were captured by the Operetta High Content Screening System (PE). The size of the spheroid (area), the fluorescence intensity, and the contrast of mCherry from individual spheroids (*n* ≥ 3) were calculated with instrument accompanying image analysis software Harmony 4.1. The resulting data were further analyzed and plotted with GraphPad Prism8 or TIBCO Spotfire.

### 2.8. Screen Potential PR Repressors Using GeCKO Library

The GeCKO library was created by the Zhang lab and consists of lentivirus constructs containing the CRISPR-Cas9 nuclease system and specific sgRNA sequences for genome-wide gene knockout in LentiCRISPR v2 backbone. Both GeCKO A and B libraries contain 3 sgRNAs per gene, as well as 1000 negative control sgRNAs. Ishikawa 12-A2 clone cells stably expressing the PR reporter gene were transduced with lentiviruses from GeCKO library A or B (at multiplicity of infection (MOI) = 0.3) to ensure most cells receive only one virus to knock-out gene across the genome. The transfectants were selected using puromycin (2 ug/mL), ganciclovir (20 uM), and hygromycin (100 ug/mL) for over one week. Using flow cytometry, high mCherry expressing cells were sorted out and seeded for colony expansion. Genomic DNA was extracted from high PR expressing clones to amplify the specific sgRNA sequence. Sanger sequence was applied to reveal the identity of the knockout genes.

### 2.9. Real-Time PCR

Quantitative real-time PCR (qPCR) was performed as previously described [24]. Comparisons of normalized expression values (ΔCt) were applied using the conventional ΔΔCt fold change method [31].

### 2.10. Western Blotting

Expression of PR, mCherry, PAEP, FOXO1, p21, Myc, cyclin D1, and β-actin were assessed by Western blotting as previously described [24,32]. Briefly, after treatment, cells were lysed using NP40 cell lysis buffer (150 mM NaCl, 50 mM Tris-HCl, pH = 7.4, 1% NP40), sonicated, and centrifuged. After running on SDS-PAGE, proteins were transferred and probed with indicated antibodies.

### 2.11. Knockout APH1A, SGPP2, SETDB1, and SOX9 in Ishikawa Cells

Three pairs of single guide RNA (sgRNA) targeting APH1A, SGPP2, SETDB1, and SOX9, or two pairs of sgRNA serving as non-target control, were synthesized and inserted into LentiCRISPR v2 vector (gift from Dr. Feng Zhang lab, Addgene plasmid #70662) by following the instructions. Sanger DNA sequence confirmed the sgRNA sequence. The lentivirus was packaged in HEK293 T cells and transfected into Ishikawa cells. The pooled stable transfected cells were selected with 2 ug/mL puromycin.

### 2.12. CRISPR-Mediated Promoter Repression of GLUD2

By applying a CRISPR-mediated gene-repression platform, catalytically inactive dCas9-fusion proteins with KRAB were used to repress transcription of GLUD2 by Lenti-EF1a-dCas9-KRAB-Puro and pgRNA-CKB.

### 2.13. Quantification and Statistical Analysis

Triplicate data are presented as the mean ±SD. All statistical analyses were conducted using Microsoft Excel. A two tailed Student’s *t*-test was used for comparisons between two groups. *p* values of less than 0.05 (*p* ≤ 0.05) were considered statistically significant.

## 3. Results

### 3.1. Construction of an Endogenous PR Reporter Gene

To insert the hygromycin resistance gene, mCherry reporter gene, at the 3′ end of *PGR*, a protospacer adjacent motif (PAM, NGG) sequence, 29 nt before the TGA stop codon, and the adjacent sgRNA before the PAM sequence, was selected. CRISPR-Cas9 genome editing technique was used to target this sgRNA to generate a double strand break (DSB) (Figure 1A). An Alt-R assay was used to confirm that the location of the double strand break was as expected (Figure 1B,C).

A donor vector containing four DNA fragments was synthesized including: (1) the 5′ end homologous recombination arm from intron 7 (488 bp); (2) the T2A (a 2A-like sequence from the insect virus *Thosea asigna*) self-cleaving peptide followed by hygromycin (1086 bp); (3) the T2A peptide followed by mCherry (765 bp); and (4) the 3′ end homologous recombination arm from exon 8 (352 bp) (Figure 1D). The donor vector was sub-cloned into the PL253 vector containing the thymidine kinase (*TK*) gene, functioning as a suicide gene to remove clones with improperly integrated reporter genes. The CRISPR-Cas9-sgRNA and the donor vector were co-transfected into two endometrial cancer cell lines, Ishikawa and ECC1 (reported as a derivative of Ishikawa cells [29,30]), to generate the PR reporter gene expressing cells (Figure 1E). Single clones containing the PR reporter gene were cultured and validated using junction PCR to ensure the accurate location of integration of the donor. Junction PCR uses primers located at: (1) the genomic DNA upstream of the recombination arm and (2) the insert gene. As shown in Figure 1E, appropriately integrated PR reporter genes were amplified in both Ishikawa and ECC1 transfected cells when assayed as a mixture of all transfectants (note the P2 and P4 lanes on Figure 1E show the correct bands. After assessing all transfectants, we selected individual clones to create the stable cell lines for further study. We were successful with both Ishikawa and ECC1 cell lines. Ishikawa clone #12 was confirmed to have the appropriately inserted PR reporter gene by demonstrating the correct insertion of the reporter gene in lanes P2 and P4. To address whether the PR reporter gene is inserted in one or both alleles, the same pair of the primers used in Figure 1B was applied to expand the genomic sequence containing the PR reporter gene fragment, as shown in Figure 1F. Compared with wildtype and the mixed population of the transfectants, which displayed 1133 bp products, only a 2984 bp band was observed for the clone #12, which indicated that both alleles contained the inserted PR reporter gene fragment (Figure 1G). Therefore, we demonstrated that the insertion might be biallelic. In a rare scenario, the PR reporter gene might be inserted monoallelic, if one of the primer binding sites was disrupted on one of the alleles. This disruption might occur when CRISPR-Cas9 generates a double strand break, or during the homologous recombination of the reporter gene DNA fragment. Even we believe this disruption only happens on rare occasions. For these experiments, Ishikawa clone 12 was used as a model to reflect Type I endometrial cancers, the classic hormone dependent type of tumor. Multiple PR reporter clones confirmed by P5 primer pairs were shown in Appendix A. A complete timeline of building a PR reporter gene and its applications was presented in Appendix A. Sequences of primers for generation and confirmation of PR reporter gene were provided in Appendix A.

### 3.2. Validating the Correlation between mCherry, Hygromycin, and PR Expression

Next, we determined that the endogenous reporter gene reflects PR expression at the mRNA and the protein levels. Romidepsin, an HDAC inhibitor known to increase PR expression through our preliminary experiments, served as a positive control to confirm the direct relationship between mCherry, hygromycin, and PR expression. Indeed, our qPCR data revealed that PR expression increased in parallel with hygromycin and mCherry expression after romidepsin treatment (Figure 2A). Romidepsin treatment increased PR mRNA 9- to 17-fold and mCherry and hygromycin 98- to 150-fold. Thus, we noted that the fold induction of mCherry and hygromycin, while reflective of an increase in PR, was higher than PR itself at the mRNA level. One potential reason for this interesting observation involves other levels of regulation to keep PR mRNA expression relatively low compared with exogenous foreign mRNAs of hygromycin resistance gene and mCherry. Figure 2B,C, Appendix A demonstrate the protein expression of mCherry, which are induced by romidepsin, substantiating the faithfulness of the new endogenous reporter vector. Since this reporter gene consistently reflects and amplifies endogenous PR expression, it provides an efficient tool to screen novel PR inducers in multiple cancer cell lines. To visualize the red fluorescence of mCherry signals, PR reporter gene expressing cells were grown in a 3D spherioid cell culture that most closely replicates the in vivo tumor growth conditions. The mCherry signal intensity was increased after romidepsin treatment (Figure 2D). Furthermore, time course experiments revealed that mCherry signals were both time-dependent and dose-dependent. The mCherry signals were observed to be at peak saturation ~72 h after romidepsin treatment (images as in Figure 2E and the time-dependent dose responses as in Figure 2F). In addition to romidepsin, other HDACi known to induce PR expression were tested. As expected, LBH589 and PXD101 also induced mCherry signals. Cells were grown in triplicate to confirm the reproducibility and stability of the mCherry signal (Figure 2G). They all have shown dose responses at 96 h after treatment (Figure 2H).

### 3.3. Screen for and Validate Potential PR Inducers

One valuable advantage of using this endogenous PR reporter gene is the ability to screen for potential PR inducers from drug libraries. Through this method, new FDA drugs that can restore functional PR expression and further sensitize endometrial cancer cells to progestin therapy can be identified. The FDA-approved 1018 drug library was applied to PR reporter gene expressing cells, followed by monitoring the mCherry signal, which represents endogenous PR expression, as previously verified. After two rounds of screening, 20 initial FDA drugs with increased mCherry signal were chosen; representative mCherry signals are shown in Figure 3A. All screening datasets were summarized in Figure 3B. Both the scatter plot and histograms of controls and screening FDA-approved drug library were presented for the hits for spheroid killing/inhibition and mCherry expression. The hit drugs with significant mCherry expression were then further tested over ten concentrations, ranging from 30 µM to 0.5 nM (example images shown in Appendix A). After conducting a proliferation assay to determine the optimal sublethal drug concentration for study (Appendix A), Ishikawa cells were tested for PR expression. CUDC-907 (a dual HDAC/PI3K inhibitor), carfilzomib (PR-171, a second-generation proteasome inhibitor), daunorubicin (a topoisomerase II inhibitor), and the positive control, romidepsin, were identified as the agents that most robustly induced PR expression (Figure 3C). Another available HDACi, AR-42 (that was not represented in the 1018 library), was also tested as a further control. In Figure 3C, the robust induction of PR is shown in response to AR-42. In order to validate that the upregulated PR was functional, the mRNA expression of well-studied PR downstream genes was evaluated after treating with the five top-picked drugs. Treatment with each drug resulted in the expected increase in total mRNA and protein levels for PRA/B, PRB, PAEP, FOXO1, and p21, while a decrease in the oncogenes Myc and cyclin D1 was observed, albeit with different efficacy. Therefore, it was determined that CUDC-907, AR-42, and Romidepsin increased functional PR expression (Figure 3D–E, Appendix A) in Ishikawa cells. To generalize this observation, two other endometrial cancer cell lines, ECC1 (reported as a derivative of Ishikawa cells [29,30]) and KLE cells, were tested and similar results were shown at Figure 3F to 3G and Appendix A for ECC1 cells, and Appendix A for KLE cells.

### 3.4. Discovery of Potential PR Repressors Using the Genome-Wide CRISPR Knockout (GeCKO) Library

To explore the novel PR repressors, we used a genome-wide gene knockout library. Our hypothesis was that knocking out PR repressors would induce PR expression. Therefore, the GeCKO (genome-scale CRISPR knockout) library was chosen; it consists of lentivirus constructs containing the CRISPR nuclease system and specific sgRNA sequences [33,34]. Since expression of hygromycin and mCherry are controlled under the PR promoter, knockout of a PR repressor will induce the expression of hygromycin resistance gene and mCherry, simultaneously. Furthermore, treatment with hygromycin selects for high hygromycin-expressing gene cells, correlated with elevated PR and mCherry expression levels (Figure 4A). As shown in Figure 4A, compared with empty Lenti CRISPR V2 infected cells, viruses from Libraries A and B both increased the percentage of mCherry positive cells in the populations: Library A induced to 5.5% and Library B induced to 7.4% (Figure 4A). Using flow cytometry, high mCherry expressing cells were sorted out and seeded for colony expansion. To date, 334 colonies have been grown and tested for PR expression. PR expression in the representative clones was shown as in Figure 4B. mRNA expression of PR was induced ranging from 3-fold to 128-fold (Figure 4B). Primer pairs for qPCR were provided in Appendix A. The genomic DNA was purified from the chosen clones, followed by PCR to amplify the sgRNA. The identity of the sgRNA was determined by Sanger sequencing. Currently, over 60 knockout genes have been identified, representing a number of gene families, including epigenetic modulators, metabolic regulators, genes associated with Notch signaling pathways, and genes with unknown functions. The potential PR repressors are APH1A, GLUD2, SOX9, SCRN1, SGPP2, TBC1D2B, BCAS4, and PHF13 (Appendix A), with additional genes are under evaluation. To validate the knockout efficacy of the targeted genes, mRNA expression of target genes was quantified in the identified clones alongside mRNA expression of PR. Indeed, the potential PR repressor genes were greatly reduced, which correlated with increased PR expression at various levels (Appendix A). Next, four target genes were chosen to validate whether they are bona fide PR repressors. APH1A, SGPP2, SETDB1, and SOX9 were found to reproducibly induce PR in knockout clones. As shown in Figure 4C, mRNA expression of APH1A, SGPP2, SETDB1, and SOX9 was decreased in the knocked-out cells using three distinct sgRNA compared with two non-target controls (Appendix A). Indeed, knockouts of APH1A, SGPP2, SETDB1, and SOX9 did result in increased PR expression, albeit with different efficiency, indicating these genes are potential PR repressors. GLUD2 is a metabolic regulator which is overexpressed in endometrial cancer and is correlated with poor overall survival, as shown in Appendix A. Increased PR expression was detected in both GLUD2 CRISPR Cas9 knockout clones, as seen in Figure 5A and GLUD2 promoter repressed cells mediated by dCas9 directed epigenetic regulation, as seen in Figure 5B and Appendix A. Compared to ECC1 cells, Ishikawa cells are more addicted to glutamine for proliferation. GLUD2 knockout Ishikawa cells are still addicted to glutamine for proliferation, as seen in Figure 5C.

## 4. Discussion

We report an innovative method to monitor the expression of an endogenous gene of interest, such as PR, and to determine novel therapeutic agents or molecular targets that can be knocked down to induce a target gene. In our example, the reporter gene mCherry and the hygromycin gene were inserted at the end of the *PGR* gene using CRISPR/Cas9 genome editing. The reporter gene consistently reflects and amplifies the endogenous PR expression in endometrial cancer cells and can be used to monitor PR levels in real-time in response to cell manipulation and treatment with therapeutic agents. The long-term goal is to enhance the discovery of novel mechanisms of carcinogenesis, as well as to test new drugs that act by controlling the expression of the target gene. Specifically, in endometrial cancer, this strategy can be used to determine new agents that can synergize with progestins for hormonal therapy.

### 4.1. Overview of Reporter Genes

The reporter gene is a well-utilized tool to monitor gene expression and activity in cell biology [35,36]. The traditional technique uses exogenous overexpression of a promoter region of the target gene and adds it to a reporter vector to measure gene expression. However, the various shortcomings of this approach have led to the development of new endogenous techniques including CRISPR, TALEN, and recombinant Adeno-Associated Virus (rAAV) [36]. A key advantage of using this endogenous reporter gene is the site-specific cut and integration of the reporter gene into the target gene; however, it is time consuming to establish it. To our knowledge, this is the first report of an endogenous PR reporter gene. Compared with other endogenous reporter genes, the unique character of this reporter gene is that it consists of two inserted genes, hygromycin for negative selection and mCherry for positive selection. Including hygromycin in the reporter gene is helpful for next step transfectant selection and is not a typical element used in the design of reporter systems. Furthermore, the incorporation of the self-cleaving peptide T2A will generate three individual proteins (PR, hygromycin, and mCherry) and prevent the formation of one large fusion protein. This ensures that these three proteins will form the correct structures and will be accurately detected. In addition, including the cell suicide and ganciclovir-sensitive *TK* gene in the donor vector ensures only correctly inserted transfectants survive [37]. We demonstrated that the PR reporter gene was likely inserted into the genomic sequence biallelic. Our unique PR reporter gene has been validated to accurately reflect and amplify PR expression, making it an ideal tool to track PR expression in cancer, development, or other research fields. In the future, we propose that mCherry could be replaced by NanoLuciferase, thereby achieving even higher signal induction. Several examples of the successful employment of the new PR reporter system are described in these studies, (1) to find new drugs that induce PR and (2) to identify the repressors of PR that themselves can be targeted to optimize PR expression. Each of these strategies has the potential to upregulate PR expression and activity as a means to improve responsiveness to progestin hormonal therapy.

### 4.2. PR Inducers

A limited number of chemicals or therapeutic agents have been reported to induce PR expression in breast and endometrial cancers. For example, histone deacetylase inhibitors (HDACi), trichostatin A (TSA), and LBH589 (Panobinostat, HDACi), as well as the hypomethylating agent and 5-aza-decitbine, were reported to induce PR expression in breast and endometrial cancer cells [21,23,38]. Of note, treatment with the Akt inhibitor MK2206 upregulated PR expression in endometrial cancer cells and in one low-grade endometrial PDX model, but not in two other high-grade PDX models, highlighting the urgent need to identify better agents to induce PR in high grade tumors where resistance to progestin hormonal therapy is most common [19,20,39].

Using HDACi to increase PR expression has been reported by our laboratory and others [23,24,25]. Consistent with previous reports, due to different binding affinities, the individual HDACi have various potencies with respect to PR induction [40,41,42]. As shown in Figure 2E, romidepsin and LBH589 more effectively increase PR compared to PXD101, another HDACi. In addition, our team identified significant improvement in progestin effectiveness when a histone deacetylase inhibitor (HDACi) is combined with progestin therapy which we termed, “molecularly enhanced hormonal therapy” [24,25,32]. Our data supported the approval of a new NIH NCTN trial, NRG-GY011. To maximize clinical interventions, we began to explore other potential PR inducers from the drug library using high throughput screening. Screening the FDA-approved drug library as a drug repurposing strategy is a goal of the New Therapeutic Uses program of the NIH National Center for Advancing Translational Science. Our studies discovered five candidate PR inducers with the potential to increase PR expression (Figure 3C). Among the five drugs, three are HDACi: CUDC-907, AR-42, and romidepsin. CUDC-907 and romidepsin are FDA approved drugs used to treat lymphoma and have also been tested to treat solid tumors in clinical trials [43,44,45]. AR-42 has been tested in breast cancer and multiple myeloma [46]. However, no studies prior to this report demonstrated an increase in PR as a therapeutic endpoint. Two additional drugs that resulted in high PR expression from our screen were carfilzomib (PR-171, a second-generation proteasome inhibitor) and daunorubicin (a topoisomerase II inhibitor). These two agents are currently used to treat multiple myeloma and leukemia, respectively [47,48,49]. This is the first report that these agents induce PR expression. The mechanism(s) by which this occurs is an important topic and a focus of our future research. However, the detailed mechanisms are warranted. Using this approach, we plan to screen more PR inducers using the four sets of drugs from the Developmental Therapeutics Program (DTP) from the National Cancer Institute (NCI): anticancer (oncology) drugs, diversity set VI drugs, mechanistic set IV drugs, and natural products set IV drugs. With these drug libraries available in hand, additional screening is on the way.

### 4.3. PR Downstream Genes

When considering how to optimize progestin hormonal therapy, it is very important to confirm that the expressed PR is transcriptionally active. To confirm PR transcriptional activity, we measured well-studied downstream genes that served as markers for PR activity. PR regulates FOXO1, glycodelin (PAEP), Myc, and Cyclin D1 through progesterone response elements (PREs) in the promoters of those target genes. In addition, PR regulates p21 and p27 through tethering on the Sp1 site in the promoters of those genes [50]. PR induces FOXO1 expression which leads to apoptosis and cell differentiation [51,52,53]. PR induced regulation of p21, p27, and glycodelin, and a decrease of cyclin D1, causing cell cycle arrest [50]. In our previous studies, we reported that PR is a negative transcriptional regulator for the oncogene Myc in endometrial cancer cells [32]. In our present studies, we discovered five potential PR inducers which not only enhance PR expression, but also affect PR downstream genes dramatically (Figure 3C), indicating that the increased PR expression is functional.

### 4.4. Reported PR Repressors

There are some reports on PR downregulation mechanisms in breast and endometrial cancer, with associated factors including G9a, JARID1A, and IGF-I. Knockout G9a and JARID1A were reported to increase PR expression in breast cancer cells and endometrial cancer cells, respectively [54,55]. IGF-I also inhibited PR expression in breast cancer cells [56].

In present studies, we have identified over 60 knockout genes, representing a number of gene families, such as SETDB1, SOX9, GDH2 (GLUD2), and APH1A. The literature and our preliminary data revealed that most of the selected genes are overexpressed in cancer cells and support tumorigenesis. The Human Protein Atlas and our preliminary data demonstrated that high expression of APH1A, GDH2, SETDB1, and SOX9 are associated with low survival in endometrial cancer.

APH1A, a component of the γ-secretase complex, was found to be a druggable cancer driver associated with high expression in endometrial and breast cancers. It is reported that increased expression of APH1A underlies endocrine therapy resistance in breast cancer [57]. We found that knockout of APH1A in clone A4-1-14 increased PR by 7-fold, and APH1A itself decreased about 6-fold (Appendix A). In a separate validation study, we confirmed that knockout APH1A can increase PR expression (Figure 4C). This is the first reporting of APH1A in relation to PR suppression.

SGPP2 is a transmembrane protein that degrades the bioactive signaling molecule sphingosine 1-phosphate. The encoded protein is induced during inflammatory responses and has been shown to be downregulated by the microRNA-31 tumor suppressor [58]. Our preliminary data demonstrated that SGPP2 mRNA was decreased about 11-fold in clone A3-21. Using CRISPR-Cas9 knockout technique, we confirmed that knockout SGPP2 can increase PR expression (Figure 4C). This is also the first reporting of SGPP2 in relation to PR suppression.

SETDB1, a histone lysine 9 methyltransferase, is markedly increased in various tumors, including ovarian, breast, lung, and endometrial cancer [59,60,61]. SETDB1 was reported to suppress androgen receptor and PR expression in T47D breast cancer cells [62]. We found that knocking out SETDB1 increased PR by 7-fold (Figure 4C). In a separate validation study, we confirmed that knockout SETDB1 can increase PR expression (Figure 4C). This is the first report of ETDB1 in relation to PR repression in endometrial cancer cells.

SOX9 overexpression has been found to play a critical role in various cancers [63]. A meta-analysis study of 3307 patients from 17 studies revealed that SOX9 overexpression predicts poor prognosis in multiple solid tumors [63]. Specifically, overexpression of SOX9 in uterine epithelial glands promotes gland hyperplasia [64,65]. Mechanistic studies reveal that SOX9 regulates Androgen Receptor (AR) expression, drives the WNT signaling pathway activation in prostate cancer, and is associated with endocrine resistance in breast cancer [66,67]. Our preliminary data demonstrated that SOX9 mRNA was decreased about 70-fold in clone A2-2-5, and PR expression increased around 4-fold by qRT-PCR (Appendix A). In a separate validation study, we confirmed that knockout SOX9 can increase PR expression (Figure 4C).

GDH2 (GLUD2) is a glutamate dehydrogenase in the mitochondria that catalyzes the reversible deamination of glutamate to α-ketoglutarate while reducing NAD(P) to NAD(P)H. It is reported that GDH accelerated proliferation of breast cancer and IDH1 mutant glioma [68,69]. It is reported that estrogen and progesterone inhibit purified GDH, suggesting that the functions of progesterone and GDH are related [70,71]. We have two GeCKO clones, A6-2-41 and A4-2-35, which both knock out GLUD2 and increase PR mRNA expression by 128-fold and 13-fold. We confirmed that both clones decreased GLUD2 mRNA expression by 3-fold (Appendix A). In a separate validation study, we confirmed that knockout GDH2 and epigenetically repressing GLUD2 expression by dCas9-KRAB (Figure 5) and guide RNA directed to GLUD2 promoter can increase PR expression (Figure 5B). Interestingly, glutamine deprivation can increase PR expression in Ishikawa cells (Figure 5B).

### 4.5. Caution in Choosing sgRNA Backbones

While we were working to identify the selected GeCKO clones, we had difficulty in distinguishing the Lenti-CRISPR-PR sgRNA v1 from the Lenti-CRISPR sgRNA v2 carried by the GeCKO library. Luckily, we found the difference between the two backbones and generated a pair of unique primers to amplify sgRNA for the target gene (v2) instead of PR sgRNA (v1). In future studies, if the GeCKO library will be used for mechanistic studies, different CRISPR backbone vectors are suggested to be chosen to generate double strand break.

### 4.6. Unbiased Method to Screen GeCKO Library

In this study, we have initially discovered many potential PR repressor genes using GeCKO library. Our technique is manually cherry-picking. The more unbiased method is using two-step PCR to amplify inserted sgRNA sequences from the pooled high mCherry-expressing cells, followed by next generation sequencing [34]. Statistical analyses will be conducted using the RNAi Gene Enrichment Ranking (RIGER) algorithm to identify candidate PR repressors. RIGER will rank the top sgRNA hits according to their enrichment. The highest-ranking genes need to be validated either using siRNA to knockdown the candidate genes or CRISPR-Cas9 technique to knockout the candidate gene.

## 5. Conclusions

Steroid hormone receptors (SHRs), such as androgen receptor (AR), estrogen receptor (ER), and progesterone receptor (PR), are transcription factors associated with the development and involvement of many cancers. Exogenous reporter genes are widely used to monitor SHR expression and activity; however, no endogenous reporter genes have been reported for these receptors. Herein, we report an innovative PR reporter gene and use it to monitor PR expression. This innovative endogenous reporter gene technique can be applied to many target genes to visualize gene expression in real-time. A few capabilities of this reporter gene include identifying novel target gene modulators with high throughput screening of the drug library and screening candidate target gene repressors or enhancers using a knockout or knockdown gene screening library. These tools allow for the real-time monitoring of target gene expression in an unbiased way and are critical for drug and mechanisms discovery in many fields. This pioneering endogenous reporter gene emerges as a new strategy to investigate gene expression and activity. This approach may provide novel and exciting breaks from bench to bed-side studies.

## Figures and Tables

**Figure 1 cancers-14-04883-f001:**
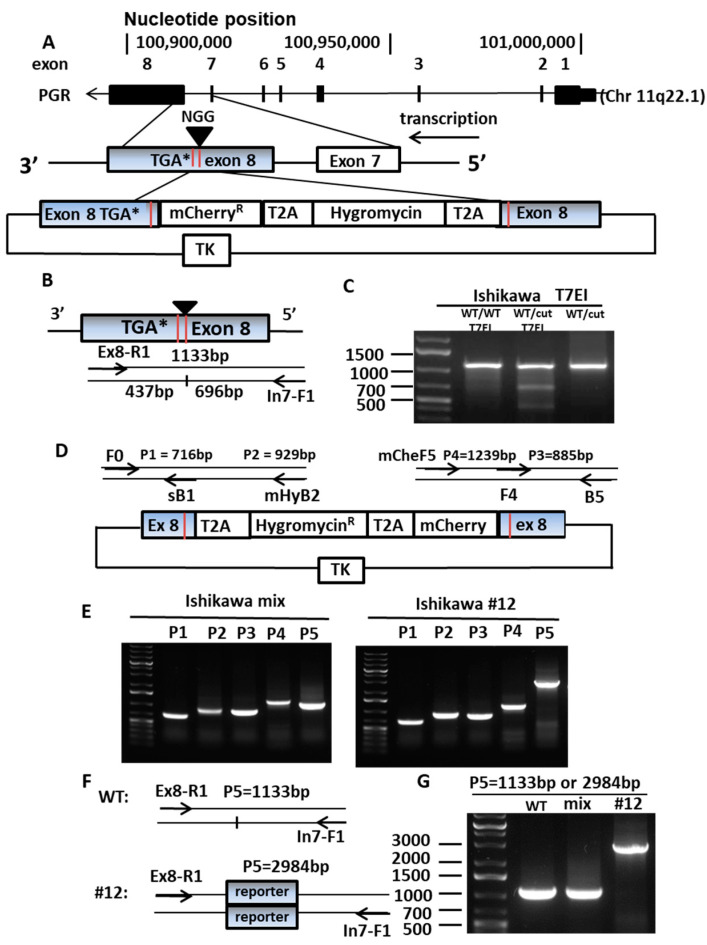
Establishing PR reporter gene expressing cells. (**A**) Construction of endogenous PR reporter gene. PGR gene map represents exon 1 to exon 8. An sgRNA was designed to target the 3′end of exon 8 to generate a double strand break (DSB) close to TGA stop codon. A donor vector was created with four components: (1) a homologous recombination arm from intron 7 to the beginning of exon 8, (2) T2A and hygromycin, (3) T2A and mCherry, and (4) ending with the homologous recombination arm at exon 8. The donor vector was inserted into the PL253 vector, including the thymidine kinase (TK) gene, to help get rid of incorrectly inserted cells. (**B**) Diagram of PCR primers for Alt-R assay to validate the location of the DSB. The DSB will generate two DNA fragments with 437 bp and 696 bp, instead of 1133 bp as the whole length of the PCR products. (**C**) Confirming the DSB in PGR exon 8 with Alt-R assay. In the presence of T7EI, sgRNA transfected cells (cut) present the two DNA fragments, 696 bp and 437 bp, contrasted by single fragment wildtype (WT). (**D**) Diagram of primers for junction PCR to validate the donor vector integrated into the correct location. Primers F0 and B5 are located outside of the recombination arm. Primers sB1 and F4 are located on the recombination arm. Primers mHyB2 and mCheF5 are located on the reporter gene. (**E**) Identify the right clone with on-site insertion of PR reporter gene using PCR. PCR products P1 (716 bp) and P3 (885 bp) serve as a positive control, P2 (929 bp) and P4 (1239 bp) stand for right insertion of the reporter gene. (**F**) Diagram of PCR primers for detection of the genomic sequence containing PR reporter gene fragment. (**G**) WT and mixed transfectants presents shorter PCR products (1133 bp). Clone #12 presents longer PCR products (2984 bp), indicating insertion of the PR reporter gene exists in both alleles.

**Figure 2 cancers-14-04883-f002:**
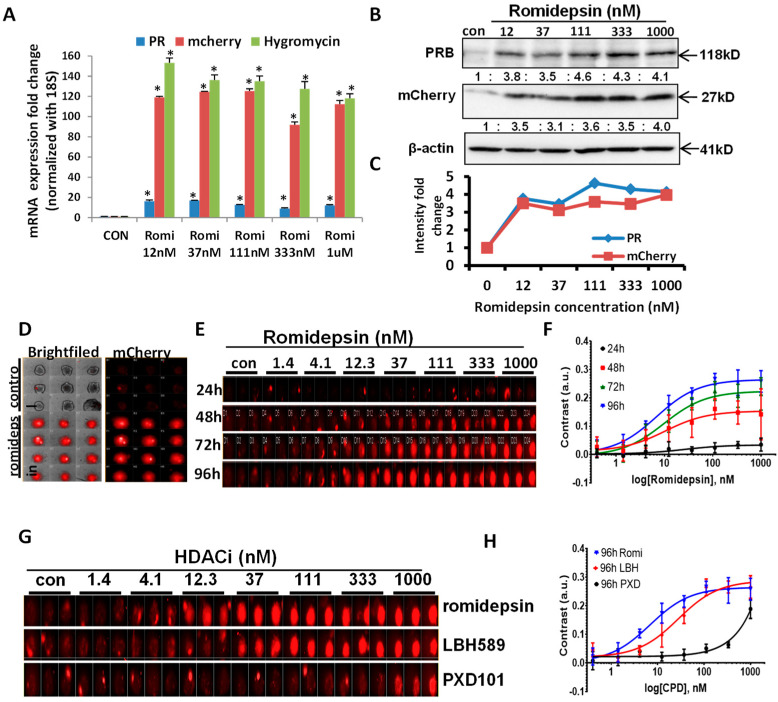
Confirming the correlation of mCherry and PR expression after HDAC inhibitor stimulation. (**A**) mRNA expression of PR, mCherry, and hygromycin were increased simultaneously upon romidepsin treatment for 24 h at 12 nM, 37 nM, 111 nM, 333 nM, and 1 µM. (**B**) Romidepsin treatment increased PR and mCherry protein expression at the indicated concentration. (**C**) Quantification of PR and mCherry fold changes in Figure 2B. (**D**) Red mCherry fluorescent signal represents PR expression after 1 µM romidepsin treatment (in triplicate) in 3D spheroid cultured Ishikawa cells for 72 h. (**E**) Dosage response (1.4 nM to1000 nM) and time course (24–96 h) experiments represent the gradually increased mCherry signal by romidepsin triplicate treatment. (**F**) The time-dependent dose responses of Romidepsin. The contrasts of mCherry fluorescence from individual spheroids were utilized for the mCherry expression increase. (**G**) Multiple HDACi triplicate treatments increased mCherry signal with the indicated dose and time course. (**H**) The dose responses of three compounds (Romidepsin, LBH589, and PXD101) at 96 h after treatment. The contrasts of mCherry fluorescence from individual spheroids were utilized for the mCherry expression increase. * *p* < 0.05 vs. control.

**Figure 3 cancers-14-04883-f003:**
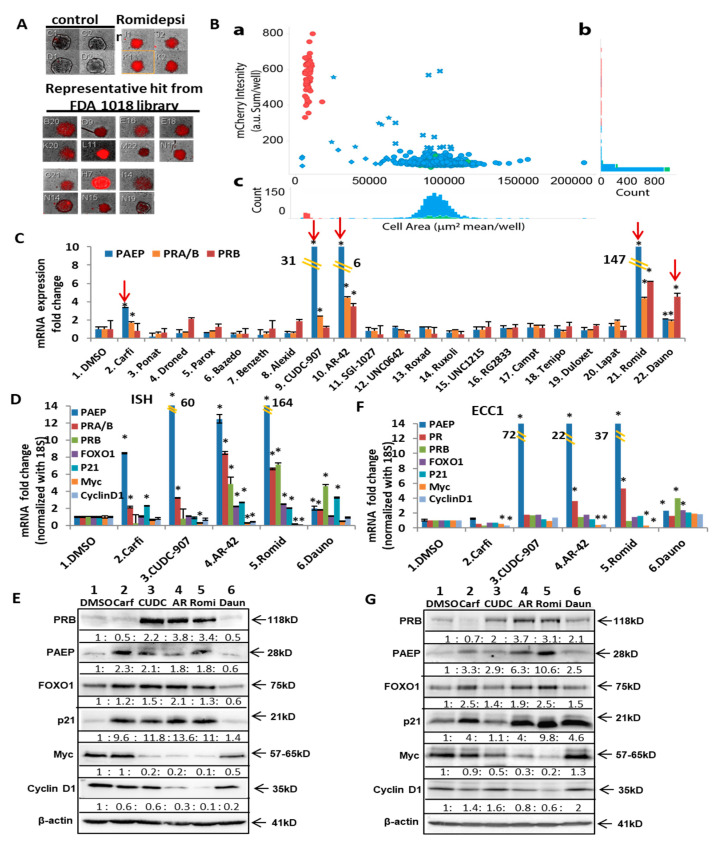
Screening and validation of potential PR inducers selected by High throughput screening of FDA approved drug library. (**A**) Representative mCherry signal in the 3D cultured PR reporter gene-expressing cells treated with FDA approved drugs at 1µM. Romidepsin (100 nM) serves as positive control. (**B**) High throughput screening of the FDA approved drug library (Selleck Chemicals, Houston, TX). **a**: Scatter plot of all compounds in the screening. Vertical axis: the mCherry intensity (arbitrary unit as sum of mCherry intensity per well). Horizontal axis: area of the spheroids (mean μm^2^/well). Green as negative control (mock treatment), Red as positive control (romidepsin at 1 μM), and Blue as the compounds from the library. ✖: Hits for mCherry intensity increase. ◆: Hits for area of the spheroid decrease. ★: Hits for both mCherry intensity increase and area of the spheroid decrease. **b**: Histogram of mCherry intensity. **c**: Histogram of area of the spheroids. (**C**) mRNA expression of PAEP, total PR (PRA/B), and PRB after being treated with the initially selected FDA drugs. Cells were treated for 24 h with Carfilzomib (PR-171) [100 nM], Ponatinib (AP24534) [500 nM], alexidine hydrochloride [100 nM], CUDC-907 [20 nM], AR-42 [500 nM], Camptothecin [100 nM], Teniposide (Vumon) [500 nM], romidepsin (20 nM), daunorubicin (100 nM), or 1 µM of other drugs. (**D**,**F**) mRNA expression of PR and its downstream genes after the treatment with the 5 top-picked drugs, Carfilzomib [100 nM], CUDC-907 (20 nM), AR-42 (500 nM), romidepsin (20 nM), and daunorubicin (100 nM) for 24 h in Ishikawa cells (E) or in ECC1 cells (G). (**E**,**G**) Western blotting of PR and its downstream genes treated with 5 top-picked drugs for 24 h in Ishikawa cells (F) or in ECC1 cells (H). * *p* < 0.05 vs. control.

**Figure 4 cancers-14-04883-f004:**
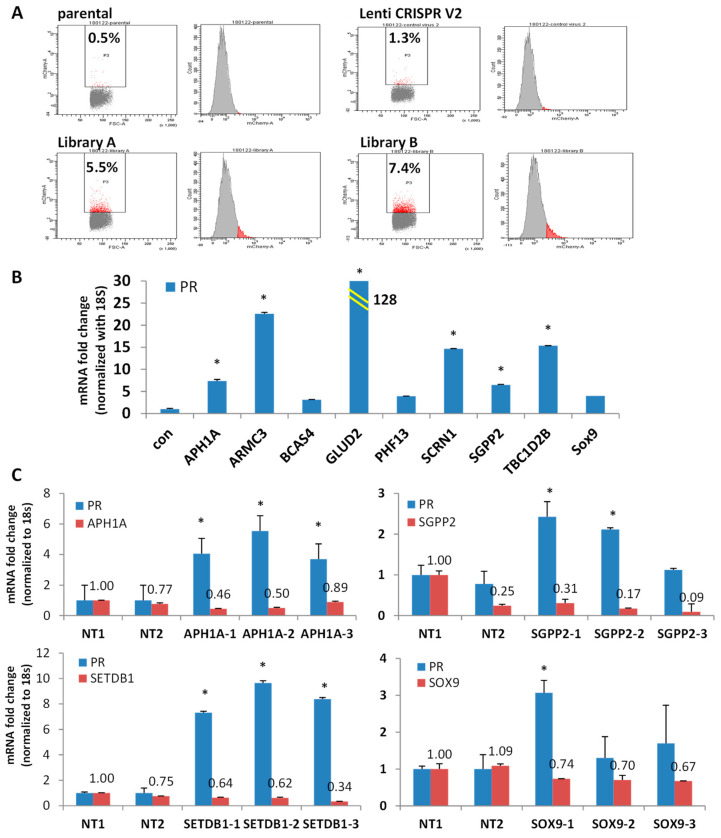
Discovery of potential PR repressors using GeCKO library. (**A**) Flow cytometry data demonstrated mCherry expressing cells (red populations) increased after transduction with GeCKO library A or B. Empty lentiCRISPR v2 vector serves as negative control. (**B**) mRNA expression of PR in the top picks of the representative clones. (**C**) mRNA expression of PR in APH1A, SGPP2, SETDB1 and SOX9 knockout clones. * *p* < 0.05 vs. control.

**Figure 5 cancers-14-04883-f005:**
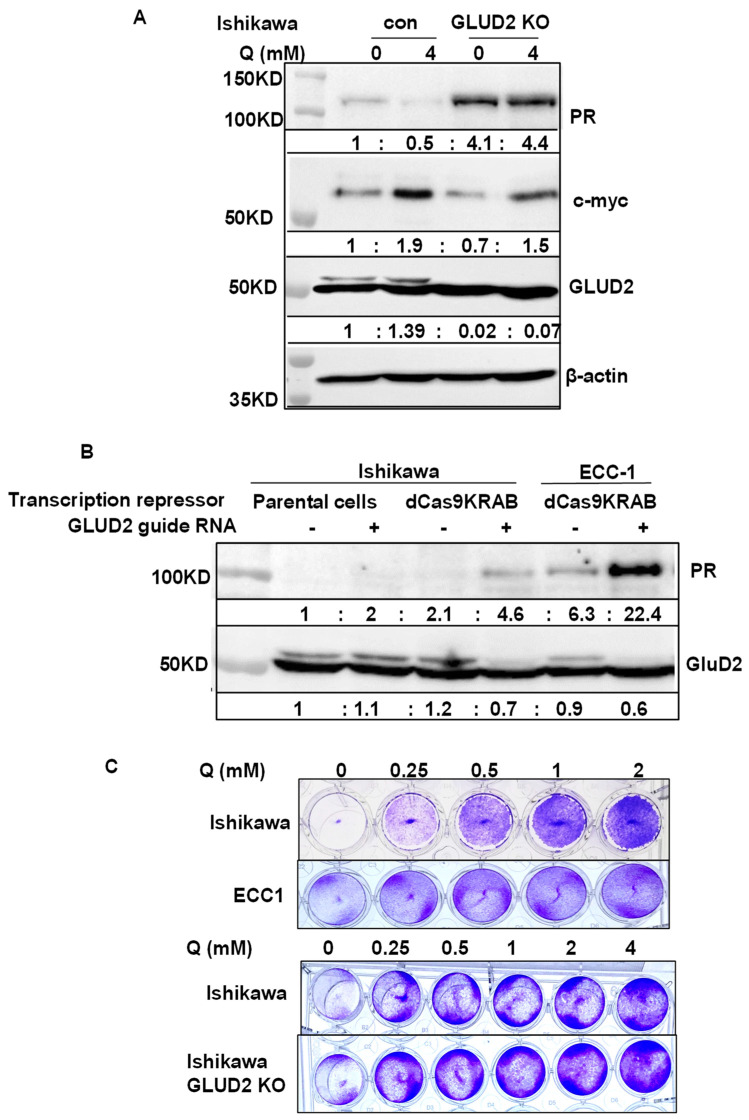
Validation PR expression regulated by GLUD2 identified from GeCKO library and GLUD2 involved glutamine. (**A**) Increased PR expression was detected in GLUD2 knockout Ishikawa cells compared to empty lentiCRISPR v2 vector infected cells. Increased PR expression was also detected in Ishikawa cells infected with empty lentiCRISPR v2 vector after 3 days’ glutamine starvation, compared to cells cultured with 4 mM glutamine. (**B**) Increased PR expression was detected in GLUD2 promoter repressed Ishikawa and ECC1 cells mediated by dCas9KRAB and GLUD2 promoter targeting giRNA. (**C**) Growth addiction to glutamine of Ishikawa, ECC1, and Ishikawa with GLUD2 knockout cells was detected by growing at the indicated concentration of Glutamine for one week.

## Data Availability

Data sharing is not applicable to this article.

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
