# Peer review of "Utilizing an Endogenous Progesterone Receptor Reporter Gene for Drug Screening and Mechanistic Study in Endometrial Cancer"

_cancers, 2022, doi:10.3390/cancers14194883_

Round 1
Reviewer 1 Report
Li et al, describe creation of a novel report system for endogenous progesterone receptor (PR) expression. This cell-line based system is useful for screening compounds that can induce PR expression.
The reporter system was created in two cell lines, ECC1 and Ishikawa. The issue is that ECC1 have been shown to be a derivative of Ishikawa 3-H-12 cells due to contamination. See PMID: 22710073 in which genotyping shows that ECC1 cells are not related to the tumor they were derived from, rather they match Ishikawa cells. The STR profiles for the two cell lines (see website and tables below) show that they are almost identical. This is a problem in the field, as ECC-1 cells continue to be used without knowledge that they are Ishikawa cells.
https://web.expasy.org/cellosaurus/CVCL_7260 (ECC1)
https://web.expasy.org/cellosaurus/CVCL_2529 (Ishikawa)
Markers: (ECC-1)
|
Amelogenin |
X |
|
CSF1PO |
11,12 |
|
D2S1338 |
20 |
|
D3S1358 |
15,17 |
|
D5S818 |
10,11 |
|
D7S820 |
9,10 |
|
D8S1179 |
12,16 |
|
D13S317 |
9,12 |
|
D16S539 |
9 |
|
D18S51 |
13,19 |
|
D19S433 |
12.2,14 |
|
D21S11 |
28 |
|
FGA |
21 |
|
Penta D |
10,11 |
|
Penta E |
11,19 |
|
TH01 |
9,10 |
|
TPOX |
8 |
|
vWA |
14,17 |
Markers: Ishikawa
|
Amelogenin |
X |
|
CSF1PO |
11,12 |
|
D2S1338 |
20 |
|
D3S1358 |
16,17 |
|
D5S818 |
10,11 |
|
D7S820 |
9,10 |
|
D8S1179 |
13,16 |
|
D13S317 |
9,12 |
|
D16S539 |
9 |
|
D18S51 |
12,19 |
|
D19S433 |
12.2,14 |
|
D21S11 |
28 |
|
FGA |
21 |
|
Penta D |
10,11 |
|
Penta E |
11,19.1 |
|
TH01 |
9,10 |
|
TPOX |
8 |
|
vWA |
14,17 |
Given this, any reference to the ECC-1 cell line should be removed. Ishikawas are notoriously unstable, so it’s possible that the “ECC-1” cells used in this manuscript are a different sub-line of Ishikawas, but further genotyping would be needed to assess this. There was a difference observed in glutamine addiction between the cell lines used in this manuscript, which may be due to them being different sub-lines of Ishikawas or due to different culture conditions.
The results should be validated in another cell line. PR-B was restored to KLEs and HEC1Bs (reference 23) using epigenetic modulators and these would be suitable cell lines for validation.
Author Response
Response: We appreciate the reviewer's concerns about rigors of scientific publication. We are aware of the ECC1 identity issue as reported by Korch et al at 2012. In a 2020 paper reported by Devor et al, genomic characterization of five commonly used endometrial cacner cell lines, including Ishikawa and ECC1 cells, were carefully assessed. From analysis on these characterization, including “histology, mutation profile, MutL homolog 1 promoter methylation, copy‑number variation, homologous recombination repair and microsatellite instability”, it was concluded that ECC1 cells were no longer exist, but this cells “are not exactly Ishikawa” and they suggest that “ECC1 to be used in vitro with this caveat in mind” [1].
Following the reviewer’s suggestions, we provide a new supplemental Fig. 3B to show Romidepsin and CUDC-907 treatment restores PR expression in KLE cells, as well as upregulation of FOXO1, PAEP and p21 expression, decrease Myc expression.
1. Devor, E.J., et al., Genomic characterization of five commonly used endometrial cancer cell lines. Int J
Oncol, 2020. 57(6): p. 1348-1357
Reviewer 2 Report
This is an exciting report highlighting an original genetic tool to measure progesterone receptor (PR) expression in real time in live cells/organoids/tumors. The construct contains mCherry/hygromycin integrated into the PR locus near the end of exon 8 which is self-cleaved through peptide sequences. In this report the authors integrate the construct though CRISPR/Cas9 editing into two endometrial cancer (EmCa) cell lines and perform a drug library screen and Gecko knockout screen to discover drugs and molecules, respectively, that alter PR expression. They identified several compounds including HDAC/proteosome/topo II inhibitors. Several molecules that putatively repress PR expression were also identified. This system has utility in tracing PR expression and deciphering mechanism of PR expression and identifying compounds that may allow progestin therapy to be efficacious. Discussion was thorough and easy to read. Some items for improvement in the manuscript are noted below:
11. I might suggest focusing the simple summary/abstract on endometrial cancer since this report is highly relevant to the treatment of EmCa and the abstract only mentions the significance of EmCa once.
22. Figure legends (except Figure 1) are not “stand alone” and are missing details. i.e. statistical tests used and drug concentrations/treatment times. Recommend adding these details.
33. How can the authors be certain the drugs given (i.e. HDACi Romidepsin) do not themselves impact c-myc expression etc? that this is not due to increased PR expression
44. The acid test would be to add progesterone to the cells re-expressing PR and test its impact on growth or the expression of PR target genes. I don’t see that a progestin was tested in any of the studies which is important to determine feasibility of induced PR expression for treatment purposes.
55. For the section on discover of PR repressors, several sentences state “hundreds of colonies” were screened or “dozens of knockout genes” were identified. Recommend adding specific numbers to the text and if necessary say additional colonies/genes are under evaluation.
66. It seems like a lot of work to knock out each candidate repressor using CRISPR – why not just test siRNA/shRNA first to rule in/out top candidates?
Minor
77. Drug concentrations/vehicle for library screen not mentioned in high content methods section
88. Fig 3B there are shapes mentioned that indicated different types of hits but these are not discernable in the figure – too small to distinguish.
Author Response
Response: We appreciate the reviewer for his/her detailed evaluation and valuable suggestions. We also appreciate that the reviewer recognizes the value of this work, the enthusiasm for our novel PR reporter gene, and the applause for our manuscript. Our response is followed point-to-point.
- I might suggest focusing the simple summary/abstract on endometrial cancer since this report is highly relevant to the treatment of EmCa and the abstract only mentions the significance of EmCa once.
Response: We agree with the reviewer and have added this sentence at the end of the abstact “ This novel endogenous PR reporter gene system facilitates the discovery of new treatment straregy to enhance PR expression and further sensitize progestin therapy in endometrial cancer.”
- Figure legends (except Figure 1) are not “stand alone” and are missing details. i.e. statistical tests used and drug concentrations/treatment times. Recommend adding these details.
Response: We apologize for the confusion. New figure legend with more details have been provided.
- How can the authors be certain the drugs given (i.e. HDACi Romidepsin) do not themselves impact c-myc expression etc? that this is not due to increased PR expression
Response: We agree with the reviewer that Myc expression is regulated not only by PR. HDACi decreases Myc expression was reported in multiple tumors as summarized in a Nature review paper [1]. Similar to Myc, other well-studied PR target genes, such as p27 and Cyclin D1 are also regulated by many different upstream genes [2-5]. Therefore, multiple PR downstream genes were tested to verify PR activity. In addition, we want to point out that our 2016 Plos one paper has established that PR is a negative transcriptional regulator of oncogene Myc in endometrial cancer [6].
- The acid test would be to add progesterone to the cells re-expressing PR and test its impact on growth or the expression of PR target genes. I don’t see that a progestin was tested in any of the studies which is important to determine feasibility of induced PR expression for treatment purposes.
Response: While it is a logical thinking, however, the fetal bovin serum (FBS) supplemented with the growth media contains progesterone. To remove progesterone (P4) or other steroid, charcoal stripped serum are needed; while in our studies, we use regular FBS which contains progesterone and it can activate PR.
We also agree with the reviewer, to test the activity of the re-expressed PR, a good strategy is adding progesterone into the cells with re-expressing PR and test its impacts on the growth and the expression of PR target genes. We are very happy to point out that our previous publications have applied this strategy to thouoghly approve that PAEP, FOXO1, p21, p27, cyclin D1, Myc are PR downstream genes, and addition of progesterone decreases cell viability. Please check these representative figures for the above conclusions.
- Fig 4B in our 2014 Oncotarget paper [7]. HDACi induced upregulation of AREG and PAEP are PR-dependent due to addition of P4 boosting PAEP transcripts.
- Fig 4A in our 2016 Plos one paper [6]. Treatment with progesterone produced a time-dependent decrease in Myc protein levels, and the addition of HDACi accentuated this effect.
- Multiple figures in 2020 American Journal of Cancer Research [8]. Fig 2 demonstrated expression of exogenous PR activates PR downstream genes in multiple tumor cell lines, which include Myc, p27, p21, Cyclin D1, FOXO1 with different efficacy. Fig 3 demonstrated that expression of exogenous PR induces cell death. Fig 5 demonstrated that progesterone promoted HDACi-induced expression of PR and its downstream genes.
- For the section on discover of PR repressors, several sentences state “hundreds of colonies” were screened or “dozens of knockout genes” were identified. Recommend adding specific numbers to the text and if necessary say additional colonies/genes are under evaluation.
Response: We are sorry for the confusion and appreciate the suggestions. We have updated with “334 colonies” and “ over 60 knockout genes” in the manuscript. We also add the sentence “ with additional genes are under evaluation”.
- It seems like a lot of work to knock out each candidate repressor using CRISPR – why not just test siRNA/shRNA first to rule in/out top candidates?
Response: We agree with the reviewer. For other candidate repressors, we will use siRNA/ShRNA to filter the top candidate. At the time of validation, we thought CRISPR-Cas9 knockout will efficient deplete the candidate genes; plus we have available CRISPR-Cas9 vectors for knockout.
Minor
- Drug concentrations/vehicle for library screen not mentioned in high content methods section
Response: Drug concentrations/vehicle for library screen has been provided.
- Fig 3B there are shapes mentioned that indicated different types of hits but these are not discernable in the figure – too small to distinguish.
Response: We are sorry for the confusion. Revised Fig. 3B has been provided.
- Falkenberg, K.J. and R.W. Johnstone, Histone deacetylases and their inhibitors in cancer, neurological diseases and immune disorders. Nat Rev Drug Discov, 2014. 13(9): p. 673-91.
- Abbastabar, M., et al., Multiple functions of p27 in cell cycle, apoptosis, epigenetic modification and transcriptional regulation for the control of cell growth: A double-edged sword protein. DNA Repair (Amst), 2018. 69: p. 63-72.
- Klein, E.A. and R.K. Assoian, Transcriptional regulation of the cyclin D1 gene at a glance. J Cell Sci, 2008. 121(Pt 23): p. 3853-7.
- Qie, S. and J.A. Diehl, Cyclin D1, cancer progression, and opportunities in cancer treatment. J Mol Med (Berl), 2016. 94(12): p. 1313-1326.
- Kress, T.R., A. Sabo, and B. Amati, MYC: connecting selective transcriptional control to global RNA production. Nat Rev Cancer, 2015. 15(10): p. 593-607.
- Kavlashvili, T., et al., Inverse Relationship between Progesterone Receptor and Myc in Endometrial Cancer. PLoS One, 2016. 11(2): p. e0148912.
- Yang, S., et al., Systematic dissection of the mechanisms underlying progesterone receptor downregulation in endometrial cancer. Oncotarget, 2014. 5(20): p. 9783-97.
- Li, Y., et al., Loss of progesterone receptor through epigenetic regulation is associated with poor prognosis in solid tumors. Am J Cancer Res, 2020. 10(6): p. 1827-1843.
Reviewer 3 Report
Thanks for reviewing interesting original study about endogenous PR reporter gene. PR association is important factor in hormone dependent cancer esp EC. If I have to ask a revision,how does author provide more information about clinically available PR IHC result (Positive/Negative)and hormone level(Blood Progesterone ) with possible PR target drug in EC management in terms of this tool mentioned in this study ?
Author Response
Response: We appreciate the reviewer for his/her encouragement. With the successful completion of national clinical trial NRG GY011 which we provided the preliminary data, we hope in the future, another promising small moluecular PR inducers will be used on clinic to treat endometrial cance patients, possible supplemented with progestin when patient blood progesterone level is low. We would expect this drug will enhance PR expression and activity, reflecting on increased PR IHC results and PR target gene expression.
Round 2
Reviewer 1 Report
This is a good study with broad interest to the endometrial cancer field. However, I don’t believe the issue with the Ishikawa and ECC-1 cell lines has been addressed appropriately. The Korch and Devor papers both perform STR analysis which undeniably shows that these are the same cell line. The Devor paper finds both to be MSI-high. Any differences found in the Devor paper are expected due to the unstable nature of MSI-high cells. But this doesn’t mean they are different cell lines. These cells are not independent of each other, therefore to use them as separate cell lines is not appropriate. The experiments are technical replicates, rather than biological replicates.
To be clear, I do not think that this requires further experiments to be performed. Only that reference to ECC-1 as a separate cell line be removed. The data could be removed from the paper, or it can be referred to as technical replication in Ishikawa cells (or a derivative of Ishikawa cells).
Author Response
Response: We agree with the reviewer’s concerns about the identity of ECC1 cells and appreciated the reviewer being patient with us. We purchased our ECC1 cells from ATCC in 2009 and got our Ishikawa cells from New York University. We don't know ECC 1 cells derived from Ishikawa cells at that time. With the reviewer’s guidance, we are willing to make the necessary changes.
As for how to make the changes, we didn’t remove the ECC1 data, while we follow the second suggestion to state that ECC1 is a derivative of Ishikawa cells. We have added this description in three places.
- On page 3, line 103. This is in the Methods section when we first introduce ECC1 cells. "ECC1, an endometrial cancer cell line, was purchased from ATCC. This cell line was reported as a derivative of Ishikawa cells [29, 30]".
- On page 6, line 206. This is in the Results section, the first paragraph when we first describe the choices of cell lines. "ECC1 (reported as a derivative of Ishikawa cells [29, 30]".
- On page 9, lines 290-291. "ECC1 (reported as a derivative of Ishikawa cells [29, 30])".
We hope we have made this issue clear, however, if we didn’t state it clearly, we humbly asked you to guide us in detail on what else we need to do to clarify it. Thank you for all your kind help.

Reviewer 2 Report
The authors have answered most of the comments sufficiently. The one experiment that would truly test the system is missing in testing plus/minus a progestin. This is a simple experiment by using charcoal stripped FBS and adding back progestin to test the endogenous reporter, as the authors acknowledge. Several previous reports were cited - however it has not been demonstrated in the current cells/system. This would greatly enhance the significance of the paper. Otherwise the reporter could be indicating nonspecific fluorescence.
Author Response
Response: We appreciate the reviewer’s intention to verify our endogenous reporter system and enhance the significance of our paper. In the first rounds of comments, the reviewer wants us to prove the PR downstream gene is PR function dependent. We have provided our evidence. In this second round, the reviewer wants us to test the endogenous reporter by adding back progestin. We feel confused what should we do to answer this question. Here are our thoughts about this issue.
- Our mCherry reporter gene is not fused with PR, because we use two T2A self-cleaving peptide between PR, hygromycin and mCherry to generate three pieces of proteins, PR, hygromycin and mCherry. We worry about the fused proteins are too big to fold correctly. Since these three genes are under the control of PR promoter, therefore, the expression of these three genes is positive correlated.
- Addition of Progesterone will specifically bind PR protein, activate PR activity and followed by proteasome degradation of PR proteins at 6 hours as reported by our group [1] and another group [2]. Progesterone won’t increase PR expression, actually it will decrease PR expression. Progesterone also won’t bind mCherry to further increase mCherry expression.
- As a principal PR inducer, estrogen receptor, when activated by estrogen, will binds PR promoter region at estrogen response element (ERE), and further activate PR transcription and increase PR expression, as we reported previously [3]. Therefore, addition of estrogen to increase PR and mCherry expression will be a strategy to validate our reporter gene system. Our pilot studies (data not shown) did suggest that adding 10nM estrogen will increase mCherry expression at 3 days and peak at 7 days. But further studies are needed.

Round 3
Reviewer 1 Report
I am satisfied with the revisions.